# Identification and Reduction of Product Carbon Footprints: Case Studies from the Austrian Automotive Supplier Industry

Kai Rüdele 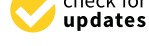 and Matthias Wolf *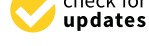

Institute of Innovation and Industrial Management, Graz University of Technology, 8010 Graz, Austria; kai.ruedele@tugraz.at
* Correspondence: matthias.wolf@tugraz.at; Tel.: +43-316-873-7796

**Abstract:** Greenhouse gas (GHG) emissions from human activities have climbed significantly above pre-pandemic levels and have reached record highs that unequivocally accelerate global warming. Industry has a significant impact on climate change, emitting at least 21% of global GHGs and making little overall progress toward its reduction until now. Reducing industry's emissions requires coordinated action along the value chains in order to promote mitigation options, such as energy and material efficiency, circular material flows, and transformative changes within production processes. This article is the first evaluation of GHG emissions generated during the manufacturing of vehicle components by Austrian companies. For this, the authors analyzed three different products of automotive suppliers according to the methodology of ISO 14067. Despite previous efforts toward an environmentally compatible fabrication, additional and significant reduction potentials were identified. These measures for product carbon footprint (PCF) reduction included the sourcing of low-carbon materials (which are already available on the market), more extensive use of renewable energy, and changes towards more resource efficient manufacturing processes and machinery. Depending on the materials used, the PCF can be reduced by up to 80%. The findings serve to prepare for future PCF reporting regulations and illustrate reduction potentials to achieve future market advantages, especially when PCFs become an awarding criterion.

**Keywords:** carbon footprint; decarbonization; car manufacturing; automotive industry; low-carbon material



## 1. Introduction

Global GHG emissions are back above pre-pandemic levels. According to the International Monetary Fund [1], manufacturing has been a particularly high contributor to recent global increases. In the meantime, energy use from industry contributes to almost a quarter of global GHG emissions, with an additional 5% coming from direct industrial processes (not including any emissions caused by freight transport) [2]. This trend, which has persisted for at least three decades, is all the more alarming as we know that climate change and global warming are mainly caused by anthropogenic emissions [3,4].

Data provided by the OECD reveal that GHG emissions from manufacturing industries and construction in developed countries have declined significantly [5]. Despite considerable economic growth and gains in prosperity, the seven largest advanced economies in the world (G7) reduced their emissions in this segment by 22% between 1990 and 2021; the member states of the European Union (EU27) were able to reduce this amount by 39% during the same period. However, these figures should be treated with skepticism as this is partly caused by a shift in the trade structure of high-income countries, i.e., domestically produced goods being substituted by imported ones [6]. As a result, emerging countries, especially China and India, have—despite their reduction of territorial emissions—multiplied their GHG emissions over the past thirty years because of increased net imports and exports of emissions. This shift in trade and the subsequent relocation of emissions from imported

goods creates the illusion that developed countries only need to reduce their territorial emissions. Attributing embedded emissions of goods to the exporting producer rather than the importing consumer (country) affects priorities and mitigation possibilities [7]. Existing and upcoming carbon tariffs such as the Carbon Border Adjustment Mechanism (CBAM) [8] will counteract this misallocation (known as 'carbon leakage') and lead to a more holistic view. Possible consequences could include changes to sourcing, up to local in-house production; the latter would lower transportation burdens and the associated emissions [6]. However, contrary effects, including an increase in global GHG emissions, may also occur [9].

Contrasting these developments, the Intergovernmental Panel on Climate Change (IPCC) is urging immediate action to tackle climate change by reducing GHG emissions and limiting global warming to 1.5 °C to 2 °C above pre-industrial levels, thus avoiding severe and irreversible destruction of the basis of human existence. Its latest assessment report emphasizes the need for deploying innovative processes and practices to enable industry's transition to net zero emissions [10]. According to the report, the most effective measures to decarbonize production in the medium term include implementing low-emission electricity and heat, energy efficiency improvements via best available technologies, fuel switching, and the substitution of high-carbon feedstock. Moreover, ecodesign, material efficiency, and waste reduction, as well as higher recycling rates up to circular economy, are the drivers for deep decarbonization trajectories [10].

Various initiatives have been launched to reduce GHG emissions, both at the national and international levels. The most ambitious set of policy initiatives comes from the European Commission's so-called Green Deal, with its declared goal of making Europe the first climate-neutral continent by 2050. This goal is legally anchored in the European Climate Law, which also sets the intermediate target of reducing net greenhouse gas emissions by at least 55% by 2030 compared with 1990 levels [11]. One featured action package is the Green Deal Industrial Plan, focusing on a simplified regulatory environment, faster access to funding, and green jobs [12]. This plan supplements previous actions of the European Union, such as the Energy Efficiency Directive 2012, the Renewable Energy Directive 2018, the Emissions Trading System (EU-ETS), and the EU taxonomy for sustainable activities. The recent report on the functioning of the European carbon market shows that in 2021, around EUR 19.4 billion (76% of total ETS auction revenues) was used for climate- and energy-related projects [13]. These investments into renewable energy, energy efficiency, and research may help the member states move closer to a net-zero emissions pathway.

Individual countries have announced their intentions to become carbon neutral within the next few decades [14,15]. For instance, Austria's federal government is pursuing the target of completely decarbonizing its energy sector and the whole of its economic system by 2040. The legislative program released in January of 2020 adds to the climate ambitions previously adopted as part of its 2018 climate and energy strategy. The new government brought the carbon neutrality target forward by a decade from 2050 to 2040 [16]. This results in the ambitious decarbonization paths that are illustrated in Figure 1.

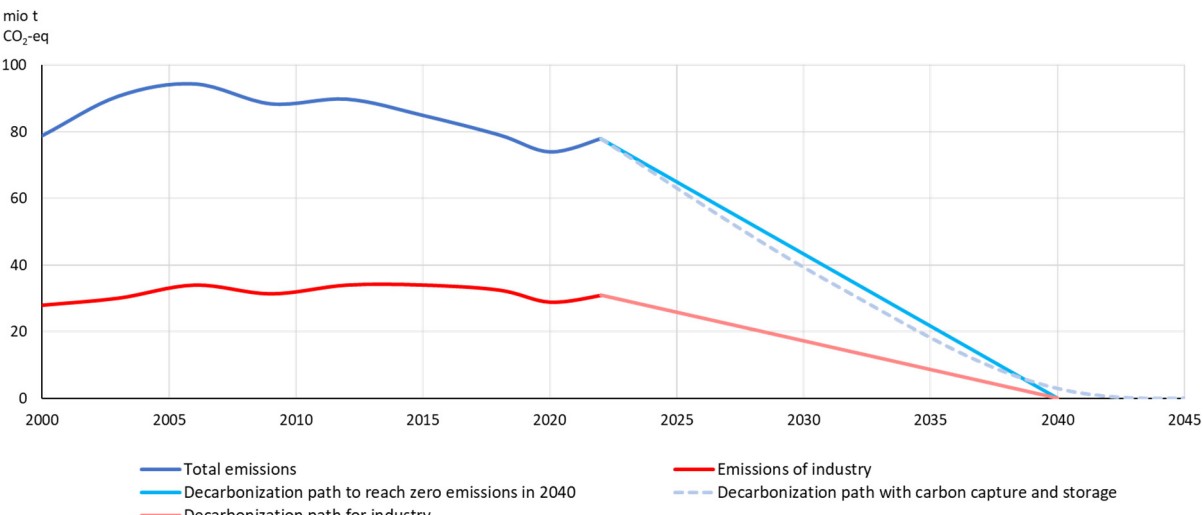

**Figure 1.** Austria's territorial annual GHG emissions and decarbonization paths (based on [5,17]).

Since peaking in 2005, Austria's total GHG emissions have declined steadily. In 2020, due to the Corona pandemic, less than 75 million tons of $CO_2$ equivalents ($CO_2$-eq) were emitted, but they started to rise again thereafter. The decrease over several years reflects the effectiveness of climate protection measures taken at that time. Occasional increases can be traced back to low prices for fossil energy and strong economic development [18]. However, considering the minimal reduction in emissions observed since 2005, a much faster transformation of the industrial sector is required in order to operate without fossil energy by 2040, thus contributing to the goal of climate neutrality. Aligned to this, an evaluation by the European Commission indicates that Austria's aimed targets for energy efficiency (especially regarding primary and final energy consumption) are lacking in ambition [19]. To achieve the 2040 target of net-zero emissions without exceeding the Austrian GHG budget of the industrial sector (see Figure 1), it is necessary that the demand for electricity, district heating, and industrial goods is reduced through behavioral changes (e.g., sharing economy, sufficiency) and measures that increase energy and resource efficiency. Additionally, the energy system must be converted to renewable energy sources [18].

A related study by Diendorfer et al. [20] determined the technical decarbonization potentials of electrification, the use of carbon-neutral gas, circular economy, and carbon capture for several sectors of Austrian industry. In the best-case scenario, these decarbonization options taken together yield an annual saving of 22 Mt $CO_2$-eq. The total investment costs associated with the implementation of these options up to the year 2040 depend on the chosen technologies and range from EUR 6.2 billion to 11.2 billion (excluding operating costs). The study focuses on energy-intensive industrial sectors (namely iron and steel production, the chemical industry, the paper industry, and the stone and glass sector) and does not examine emission drivers and approaches to cut GHG emissions for other sectors. Avoiding emissions through the combustion of fossil fuels is the central point, whereas the reduction of embodied carbon emissions by material efficiency and substitution is subsidiary.

Furthermore, the issue of sustainability is gathering momentum in the automotive industry. Along with both a shortage of resources and the standards and regulations put in place to penalize pollution, heightened environmental awareness on the part of customers may also push the automotive industry toward more ecological production [21]. The potential is available in that GHG emissions from car production of over 7 kg of $CO_2$-eq per kg of vehicle weight can be more than halved through renewable energy and sustainable materials [22].

Regarding the aforementioned recent developments, a rigorous change is required that will focus on the product's life cycle, including all embodied emissions, irrespective of

their origin. A life cycle assessment (LCA) of product–production systems helps to identify potentials to reduce environmental impacts and evaluate viable solutions and alternatives. In engineering science, the LCA has become established not only as a sustainability assessment tool but also as a research methodology [23,24]. Like other quantitative approaches, an LCA can assist or improve a decision-making process and is beneficial for operational research [25,26].

The carbon footprint is a study based on a broader LCA and expresses the quantity of GHGs (measured in kg $CO_2$-eq) emitted into the atmosphere by an individual, organization, process, product, or event from within a specified boundary [27]. If the scope of the study starts with the extraction of raw materials ('Cradle-to-x') and considers all upstream activities, the embedded emissions of imported goods become transparent.

This paper presents the authors' approach to determine the product carbon footprint (PCF) of three components produced by Austrian automotive suppliers: a crossmember, a rear axle carrier, and a grill shutter actuator. By compiling measured and calculated inputs and outputs, data were generated from which the amount of released GHG was determined. To our knowledge, this study is the first of its kind considering industrial goods manufactured in Austria alongside associated characteristics such as electricity mix, supply chains, and process maturity. It reports the main challenges and limitations of data collection and analysis but also provides recommendations for a structured procedure, since there are more than 60 initiatives and methods for calculating and reporting PCFs [28]. The results are used to identify the largest emission drivers and to quantify potential savings.

This article is structured as follows: Section 2 describes the main characteristics of the (Austrian) automotive industry, primarily with regard to economic and environmental aspects. Section 3 summarizes and reviews relevant PCF studies in the automotive sector, while Section 4 presents and discusses the empirical findings from Austrian automotive suppliers. Finally, Section 5 includes the conclusion and suggestions for further research.

## 2. Characteristics of the Automotive Industry

### 2.1. Economic Relevance

With regard to revenue and resource consumption, the automotive industry is one of the world's most important economic sectors. In the last five years, the world's annual motor vehicle production, comprising both passenger cars and commercial vehicles, ranged between 78 and 96 million units [29]. More detailed insights on the six largest vehicle producing countries, as well as Austria, are provided in Table 1. The market volume of global automobile production amounts to USD 3 trillion per year [30] and is expected to reach USD 3.8 trillion by 2030 [31]. The European automotive sector employs 13 million people directly and indirectly, accounting for 7% of all EU jobs and 11.5% of all EU manufacturing jobs [32].

**Table 1.** Production statistics from 2022 in 1000 units, with the previous year's figures in brackets [29].

| Country | Passenger Cars | Commercial Vehicles | Total Production | Change Compared with 2021 | Share of Global Production |
|---|---|---|---|---|---|
| Austria | 108 (125) | 0 (12) | 108 (137) | −21% | 0.1% |
| China | 23,836 (21,408) | 3185 (4674) | 27,021 (26,082) | 3% | 31.8% |
| Germany | 3480 (3096) | 197 (213) | 3678 (3309) | 11% | 4.3% |
| India | 4439 (3631) | 1018 (768) | 5457 (4399) | 24% | 6.4% |
| Japan | 6566 (6619) | 1269 (1228) | 7836 (7847) | 0% | 9.2% |
| South Korea | 3438 (3163) | 319 (300) | 3757 (3462) | 9% | 4.4% |
| United States | 1752 (1563) | 8309 (7604) | 10,060 (9167) | 10% | 11.8% |
| World | 61,599 (57,054) | 23,418 (23,092) | 85,017 (80,146) | 6% | |

Although not as substantial as in other European countries [33,34], the automotive (component) industry has long been an important pillar of the Austrian economy. According to Statistik Austria [35], it directly employs at least 37,000 people and accounts for over

8% of the total revenue and 6% of the gross value added (GVA) of the total manufacturing sector (ISIC section C). In 2022, the GVA of the Austrian automotive industry (C 29 ISIC) amounted to USD 3.1 billion, which was equal to one percent of the national GVA [36]. In addition to the production of passenger cars, motor vehicle components (1.6 million engines and transmissions in 2021) and motorcycles (179,000 in 2021) are also manufactured in Austria [37].

### 2.2. Current Environmental Impact of Automobile Production

Even though Austrian automotive companies only emit a fraction (154,000 t $CO_2$-eq) of what other industries (e.g., manufacturing of machinery: 430,000 t $CO_2$-eq, [38]) emit, businesses in the transportation sector are particularly challenged to reduce their emissions, as road vehicles alone account for 16% of global energy-related $CO_2$ emissions [39]. New technologies and services, regulations, and usage patterns of vehicles have already transformed or even disrupted the auto industry [40]. Consequently, environmentally sound production processes represent an additional challenge for this industry.

By definition, the automotive sector does not belong to the energy-intensive industries [41–43]; however, its sourcing activities in those industries are significant. The European Steel Association estimates that the automotive sector accounts for 17% of steel demand in the EU, especially strip mill products [44]. On average, 900 kg of steel is used per passenger car, mainly in the body structure, drive train, panels, doors, and suspension [45]. With a share of 18%, the manufacture of vehicles is also one of the main industrial consumers of aluminum globally [46]. In 2022, the total aluminum gross demand for the European automotive industry amounted to more than 2.8 million tons, of which nearly half was processed through energy-intensive casting [47]. With a production volume of 11 million tons in 2022, the flat glass sector represents the second largest sector of the European glass industry [48], of which almost 15% is processed into glazing for the automotive and transport industry [49]. As Sato and Nakata [50] demonstrate, a large quantity of copper ore is also needed for vehicle production. About 6 to 18% of a vehicle's curb weight is made up of plastic, whereas rubber contributes to approximately 4 to 7% [50–53]. Due to their tendency to have a lower global warming potential, (synthetic) polymers are less relevant than metals (e.g., [54,55]).

Damert and Baumgartner [56] found that final vehicle manufacturers (original equipment manufacturers (OEMs)) are more ambitious in carbon compensation than their suppliers due to their greater exposure to public attention and associated stakeholder pressure. Because of their size, they also have better access to financial resources and knowledge. While OEM companies often come under scrutiny because they are the main party held responsible for tailpipe and fleet emissions, car part manufacturers play a key role in cutting emissions caused by manufacturing. Currently, there are few first-tier suppliers delivering complete systems and modules, whereby they expanded their capabilities, dominate the technologies of a respective sector, and contribute to innovation significantly [57]. However, the changeover to low-carbon products and production was not rewarding until recently due to a lack of demand and high investments [58]. Nevertheless, there are numerous concepts to reduce supply chain emissions by involving (sub)suppliers and multiple tiers, such as low-carbon procurement or product stewardships (e.g., [59,60]). Already, more than twelve years ago, Lee [61] demonstrated how OEMs could work with their suppliers to improve the environmental performance of manufacturing processes by measuring the PCF of single components. An index proposed by Azevedo and Barros [62] showed that (environmental) sustainability of the UK's automotive supply chain has improved in recent years.

There are also characteristics of the automotive industry that come with positive effects for the environment. On the one hand, it is operating under considerable cost pressure, and the predominate sector adopting lean principles results in an efficient resource consumption [63,64]. On the other hand, the automotive industry is characterized by high volumes and automation, resulting in corresponding economies of scale. One-off

productions and their associated resource-intensive and polluting manufacturing processes, such as lost-wax casting [65], are the absolute exception.

### 2.3. Trends and Challenges in the Automotive Sector

Forecasts predict that vehicle production and sales will remain above current levels (at least until 2026) and that SUVs will remain the most popular segment [66]. Vehicles are also becoming larger and heavier (e.g., [67]). Estimates assumed that this resulted in the additional consumption of up to 9.2 million tons of steel in 2022 [68]. With the mass market transition to electric vehicles and the accompanying lighter-weight construction, the average aluminum content per car manufactured in Europe is expected to increase from today's 205 kg to 256 kg by 2030 [47]. By 2050, the automotive industry's demand for aluminum is even predicted to double [46]. But each kilogram of aluminum that substitutes steel in automotive applications saves up to 20 kg of $CO_2$-eq over the life of the vehicle [69]. In the future, emissions from production will have a higher share than those from the entire use phase. According to the World Economic Forum [53], 60% of automotive life cycle emissions will come from materials by 2040. Both aluminum and steel are highly recyclable, however there are partly contradictory statements regarding the extent to which this potential is exploited. While the European Aluminium Association [70] states recycling rates reach over 90% in the automotive sector, the International Aluminium Institute [71] forecasts that, even by 2050, primary aluminum will still constitute approximately 45% of the aluminum used by the automobile industry. According to the American Iron and Steel Institute [72], the steel industry is recycling nearly 100% of the steel in automobiles by the end of their life cycles. A large number of studies encourage vehicle production to intensify the use of recyclates, selective disassembly to single out hazardous or valuable components, and separation processes to liberate desired materials to save energy, costs, and emissions (e.g., [51,73–75]). Greenpeace demands a transition to zero-carbon steel [68].

### 2.4. Actions of the Automotive Industry

On their websites, nearly all vehicle manufacturers publish extensive sustainability reports that examine not only current and future fleet consumption but also the environmental burden of vehicle production. In addition to actual performance indicators, the reports also include targets and measures for decarbonization and resource conservation. For European car manufacturers, such reporting has already been firmly established for more than 10 years [76], and Lukin et al. [77] cursorily examined the extent to which leading automotive companies meet certain Sustainable Development Goals (SDG) of the United Nations. Meanwhile, most sustainability reports are following standards provided by the Global Reporting Initiative (GRI) [78]. A GRI compliant sustainability report must contain a section focusing on used materials and supply chain-related sustainability issues (especially 'procurement practices' and 'supplier environmental assessment') [79]. An analysis by Caliskan et al. [80] suggested that sustainability reports from automotive suppliers tended to have a weaker stance against SDGs than those of OEMs.

OEMs use these reports to announce their own milestones. BMW [81], for example, is striving for the 'lowest possible level of resource consumption in production', while Toyota [82] wants to 'reduce $CO_2$ emissions from global plants by 30% compared to 2013 levels' and 'achieve carbon neutrality at all global plants by 2035'. Such ambitious plans present competitive opportunities for suppliers to aid OEMs with their targets. An overview regarding low-carbon automotive steel partnerships is provided by Liu et al. [68].

OEMs are perceived as more (pro)active than suppliers when it comes to implementing climate change measures [56] as they focus more on carbon compensation measures and engagement with policymakers to legitimize their business operations. Damert and Baumgartner [56] also recommend that OEMs should use their buying power to encourage their suppliers to take more ambitious actions on climate change.

The initiation of low-carbon products can start with carbon-accounting and monitoring activities that evaluate the GHG emissions caused by suppliers (e.g., [83]). As long as

20 years ago, the German Association of the Automotive Industry (VDA) introduced the first version of its LCA Data Collection Format as a means for collecting, processing, and documenting relevant data to enable automobile manufacturers and their partners along the supply chain to environmentally improve their products. The related documents shall be applied in order to use the obtained data to evaluate (design) options and justify environment-related decisions concerning the production of automotive components and operating materials for automobiles, as well as processes used for production, recovery, and disposal. The main purpose is to simplify and standardize the compilation and quantification of inputs and outputs for a product (life cycle inventory, LCI) for both data suppliers and users [84,85].

## 3. Previous Studies to Determine Carbon Footprints in the Automotive Supply Industry

All studies known to the authors show that the upstream carbon footprint accounts for a substantial proportion (at least 75%) of the overall climate impact of motor vehicle manufacturing (e.g., [86,87]). This coincides with the sustainability reports from OEMs. The GHG emissions from the production of suppliers (Scope 3 category 'purchased goods' according to GHG Protocol, [88]) surpass those of manufacturing and the final assembly by an OEM by several times. Table 2 compares the information of eight of the ten OEMs with the highest number of sales according to Forbes [89]. The two missing companies either did not publish a suitable report (SAIC Motor) or the data did not allow for a breakdown of the automotive business (Honda Motor). Unless otherwise stated, the values related to all brands of the respective group. If no data per produced vehicle were reported, the total emissions of the particular scope were divided by the production volume. Low values for 'Logistics' suggest that only upstream or downstream transportation and distribution were reported.

**Table 2.** 2022 GHG emissions in t $CO_2$-eq per vehicle.

| Manufacturer (Group) | Supply Chain (Purchased Goods) | Production by OEM | Logistics | Source |
|---|---|---|---|---|
| BMW | 8.3 | 0.3 | 1.0 | [90] |
| Ford | 9.5 | 0.7 | 1.8 | [91] |
| General Motors | 8.1 | 0.7 | 2.4 | [92] |
| Hyundai | 5.0 | 0.6 | 0.3 | [93] |
| Mercedes-Benz Group | 8.7 | 0.3 | 1.1 | [94] |
| Stellantis | 6.7 | 0.6 | 0.2 | [95] |
| Toyota | 10.5 | 0.8 | 0.6 | [82] |
| Volkswagen | 9.3 | 0.7 | 0.5 | [96] |

Annotations: Hyundai w/o Kia; Toyota data from 2021 and w/o Daihatsu; Volkswagen limited to passenger cars and light commercial vehicles of all brands.

In addition to OEMs and suppliers' own disclosures, there are numerous publications in the field of LCA, both at the overall vehicle level and for individual components. The analyses of Sullivan et al. [52] and Sato and Nakata [50] used data from numerous sources to estimate material composition, determine involved processes, and calculate the cumulative energy consumption of car production. While Sato and Nakata [50] only detailed resource consumption and did not consider the environmental burdens, Sullivan et al.'s work pointed out related $CO_2$ emissions (but not the total GHG-equivalent emissions) [52]. Gebler et al. [97] not only compared already known production-related GHG emissions but also analyzed the relative influence of annual production volume on the resource demands of an automotive factory. The findings appeared to be consistent with Sullivan et al., wherein about 40% of the energy consumption and carbon emissions for part manufacturing and vehicle assembly were fixed and resulted from the plant's base load and production readiness, which included lighting, heat, ventilation, air conditioning (HVAC), and material

handling [52]. Rivera and Reyes-Carrillo [98] had already shown that over 80% of the hazardous emissions from automobile manufacturing were associated with the painting stage, and their work also provided guidance to assess potential solutions. Finally, Kim and Wallington [99] discovered that the materials used in lightweight vehicles were more carbon intensive to extract and manufacture in many cases.

Nearly the same applies for suppliers and vendor parts: in their case study, Cecchel et al. [100] assessed a suspension crossbeam for commercial vehicles, concluding that the extraction of primary aluminum was by far the most energy-intensive stage in the supply chain, followed by die casting. The situation was similar in a case study that calculated the energy demand and GHG emissions of front subframes, which showed that the material production predominated the parts manufacturing in all scenarios [101]. For a rear crash management system, Del Pero [102,103] revealed that, independent of the selected metal, the raw material extraction and production caused more GHG emissions than the subsequent manufacturing processes. Basically, the same module has been the subject of other publications, namely Grenz et al. [104] and Ostermann et al. [105]. The latter analyzed a variant containing carbon fiber-reinforced plastics, but even then, the input materials still had a greater impact on the carbon footprint than the production of the final component (via casting, forming, joining, coating, etc.). Delogu et al. [106] discussed the environmental impact of two different composites suitable for automotive dashboard panels. In all categories, with regard to GHG emissions, the raw materials phase had a higher impact than the manufacturing phase, and raw material extraction caused about three times as much emission as further processing. This was similar to car floor pans, where every kg $CO_2$-eq that arose in manufacturing was accompanied by six kg $CO_2$-eq during resource extraction [107]. Moreover, a comparison of different material compositions of vehicle's body and chassis by Raugei et al. [108] showed that raw materials influenced GHG emissions much more than manufacturing.

One of the most widely analyzed components in scientific publications are batteries, especially those used in electric vehicles (an overview of previous LCAs on this subject can be found in Dieterle et al. [109]; Erakca et al. [110]; and Li et al. [111]). However, few studies make a clear distinction between the emission sources along the manufacturing chain. Ellingsen et al. [112] outlined the variations found within the published studies of the time and reported on different contributions from raw material acquisition, manufacturing, and assembly. Only in 4 out of the 14 detailed carbon footprints, cell manufacture and pack assembly together yielded GHG emissions equal to or greater than material extraction. More recent studies such as Dai et al. [113] and Kelly et al. [114] concluded that the collective upstream production of battery materials used much more process energy, respectively emitting more GHGs than the cell production and assembly process.

This was contrasted by a study by Li et al. [115], who proved that the GHG emissions caused by the production of raw materials for all components of a diesel engine were only one to two percent higher than the emissions from manufacturing. The only case where this ratio was reversed was provided by Lee [61]. He examined a bumper where the raw material production accounted for 18% of the total product carbon footprint, whereas the in-house manufacturing counted for 70% and distribution counted for 12%.

Further studies on vehicle components, such as gasoline engines [116], ignition coils [117], or door panels [118], however, have provided only limited information about the contribution of individual production stages to GHG emissions. Nevertheless, a noticeable number of publications related to the automotive industry used LCAs to compare different systems (e.g., [119]), designs (e.g., [118,120,121]), materials (e.g., [122–128]), production techniques and machine settings (e.g., [129–131]), and/or production sites (e.g., [111,124]). These publications, in turn, confirmed the observations of Böttcher and Müller [58] regarding low-carbon operations.

Moreover, there is a substantial body of research dealing with the evaluation of recycled or bio-based materials for automotive applications (e.g., [132–137]). Similarly,

there are LCA studies on whether additive manufactured automotive components offer opportunities to reduce environmental impacts (e.g., [138,139]).

The authors are not aware of any openly accessible LCAs of vehicles or automotive components manufactured in Austria. Existing studies mainly compare different propulsion technologies in which the Austrian electric energy mix is considered for the use phase of electric vehicles (e.g., [22,140]). Additionally, Diendorfer et al. [20] and Meyer et al. [141] did not quantify GHG emissions on a product level for the Austrian automotive (supply) industry. The LCA for a side impact beam, presented by Mair-Bauernfeind et al. [142], is a prospective yet hypothetical scenario in which Austria is considered as a potential production site; however, this has no relevance due to a lack of real data. LCA-related results for Austria listed by Ladenika et al. [143] are also not directly related to the automotive industry.

In summary, most, but not all, studies to date suggest that, also at the component level, upstream activities contribute more to PCFs than the processing at the supplier's site. However, automotive-related manufacturing in Austria has not been studied in detail until now. Therefore, we analyzed three different car components finished in Austria to incorporate country-specific conditions to close this research gap.

## 4. Case Studies of Three Automotive Components

### 4.1. Common Methodology and Set-Up

The main objective of our analyses was to assess the carbon footprint associated with the production of the respective component. ISO 14067 [144] was chosen as the cornerstone standard for how to quantify the GHG emissions; thus, all PCF studies were subdivided into four interrelated phases: goal and scope definition, inventory analysis (LCI), impact assessment, and interpretation. Overall, our studies drew on twelve months of close industry cooperation with two companies from the automotive supply industry, incorporating data provided by the companies (e.g., measured energy and material consumption on machinery level, production volumes, and sales) and generated by the researchers (e.g., calculations related to transport and waste) through several shop floor visits, interviews with company contact persons and experts, and correspondence with external stakeholders along the supply chain (e.g., forwarding agents).

For all cases, the functional unit was defined as one salable unit of the considered component fabricated in Austria. All analyses were carried out cradle-to-gate, i.e., from the exploration of the raw materials to the production of the individual component for the OEM. Those steps carried out in between by the sub-suppliers and forwarding agents, as well as the treatment of waste generated during manufacturing, were also considered. As requested by VDA [85], general efforts such as the lighting and heating of the halls were also incorporated. Downstream transportation and distribution were ignored; in Cases 1 and 2, the OEM was responsible for transporting finished components. The product from Case 3 was supplied to different companies (first-tier suppliers and OEMs) so that no general destination could be determined. Also, the use stage, including use profiles and end-of-life treatment of the product, was set as irrelevant. Resource consumption needed for product development ('Engineering'), capital goods, business travel, and employee commuting was excluded because no purposive recording and allocation method was possible. Where unavoidable, allocation methods were used that either reflected an underlying physical relationship or were grounded on the proportion of the economic value (usually a share of the revenue) of the products. In accordance with ISO 14067 [144], carbon offsetting, which was also especially evident in the purchased electricity mix, was excluded. Figure 2 shows the generic product system for the cases (there were slight differences in individual cases since only Case 3 had an assembly). Elements shown in grey could not be assessed and were therefore outside the system boundary.

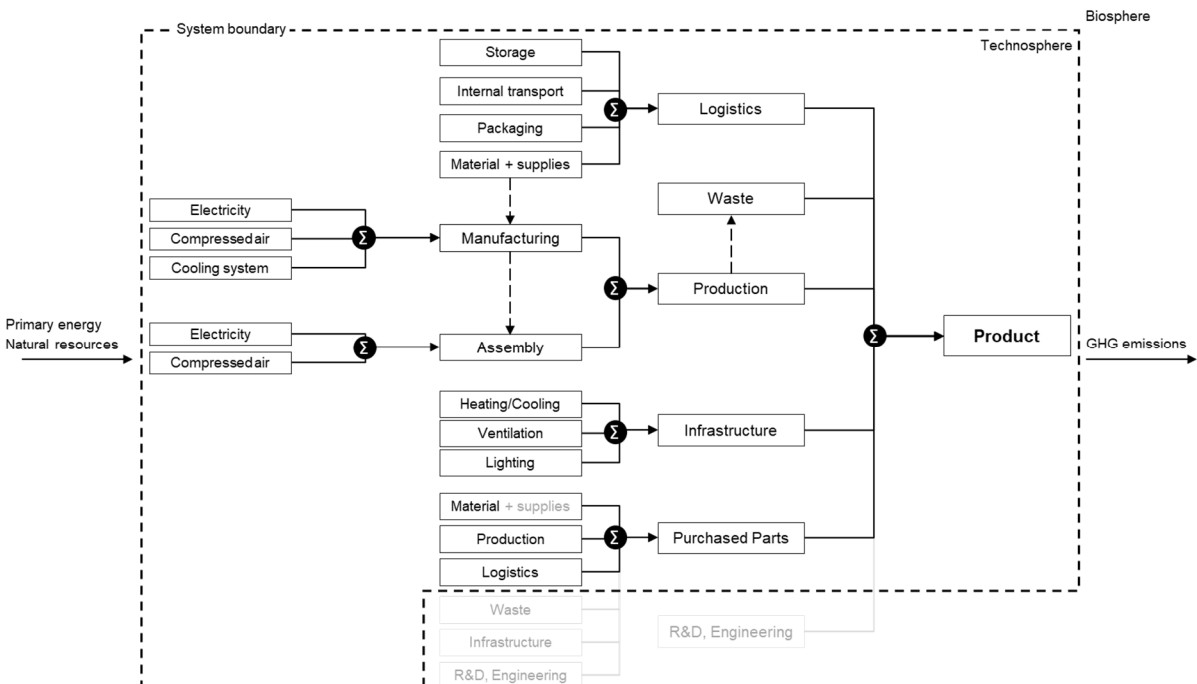

**Figure 2.** Product system used for PCF studies.

The product life cycle was excluded, as the focus was on production-related impacts. The system boundary describes the biosphere–technosphere relationship, where the elementary flows (primary energy and natural resources) represent necessary inputs to the technical system. The technosphere contains processes that ultimately generate emissions that migrate into the biosphere.

In the LCI phase, data were collected and validated. For this purpose, production data of the focal supplier was collected by measuring all kinds of energy and material consumption. The considered time period was the twelve months before the start of each study and was chosen to smooth out fluctuations in energy consumption (caused by the heating season). Sub-suppliers were asked to, at a minimum, fill out the input/output table ('Data Collection Format Spreadsheet') provided by VDA [85]. Transports were calculated based on information provided by forwarding agents. Resource consumption for defective products and parts needed for tests (both to go unsold) was tracked and attributed to the PCF of saleable products.

Using the product specifications, all raw materials could be traced back to their point of extraction, mainly by using data from the GaBi v2022.1 database [145]. The regionally specific datasets contain associated inputs from nature and emissions, including estimations on losses. In that respect, cut-offs could be avoided as far as possible. Materials not found in either the GaBi database or the other available databases (esp. Ecoinvent v3.8) were modeled from information available in the databases, or they were replaced by similar materials.

The subsequent impact assessment of the PCF study was used to calculate the potential climate change impact by multiplying the inventoried values by their substance- or resource-specific conversion factor. In this way, all GHG emissions inventoried in the LCI were weighted in terms of their impact intensity relative to carbon dioxide (expressed as kg $CO_2$-eq). With regard to the time horizon, ISO 14067 [144] follows the advice of IPCC, which is why all the following figures refer to 100-year global warming potential.

We used the concluding interpretation to screen materials and processes to identify significant sources of GHG emissions and perform a sensitivity analysis of relevant inputs. This led to recommendations for action, mainly from an ecological perspective.

### 4.2. Case Study 1: Crossmember

A crossmember is a structural component connected to the vehicle's body at different points that ensures its stability and stiffness (torsion resistance). The crossmember assessed in this case is comparable to the innovative variant (for which an LCA is also available) mentioned in Maltese et al. [121], but it is simpler in design and easier to manufacture.

The crossmember was manufactured in a single production step through the use of a transfer press (Figure 3). This press was also used to manufacture other parts, which was appropriately reflected by allocation. The component was made entirely out of aluminum sheet and there was an agreement between the OEM and the supplier to use aluminum with at least ten percent recycled content. Furthermore, there were conservative specifications from the OEM on how to calculate a PCF. In addition to the 'polluter pay principle', reselling scrap metal was counted as waste disposal, even if it was recycled.

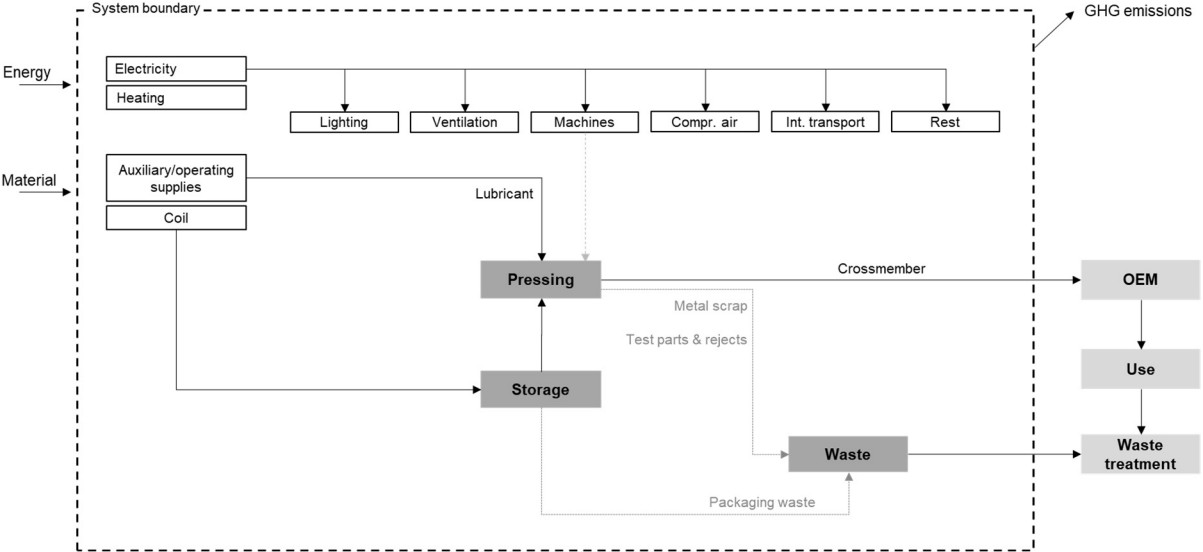

**Figure 3.** Flow chart for crossmember production.

Around 30% of the aluminum used came from press remnants and was sold to a recycler. Lubricants directly applied onto the coil material were recorded, along with the (proportional) circulation lubrication of the press.

Besides the directly attributable consumption of the machine, there was also overhead consumption that arose from the infrastructure. Energy consumption for HVAC was determined based on the floor space and the throughput time required for production. Energy consumption for lighting was determined in a similar way (via the number of lamps in the related production area and the time needed to produce one component). Revenue was chosen on an apportionment basis for other aspects such as ancillary areas (e.g., offices, the canteen, and sanitary facilities) or the energy demand of jointly used forklifts and conveyors.

Data on the transport of coils and scrap metal (esp. by means of transportation, distances, frequency, and utilization), as well as internal and external warehousing (proportional energy consumption determined by the required space and length of the storage period), were also gathered.

As illustrated in Figure 4a, 97% of GHG emissions were linked to the raw material used and only 1% of emissions were generated in the production of the crossmember. The operation of the machine (including cooling) had the highest energy consumption, followed by heating. Since the aluminum coils were sourced from Scandinavia, transport contributed more to the PCF than the subsequent processing.

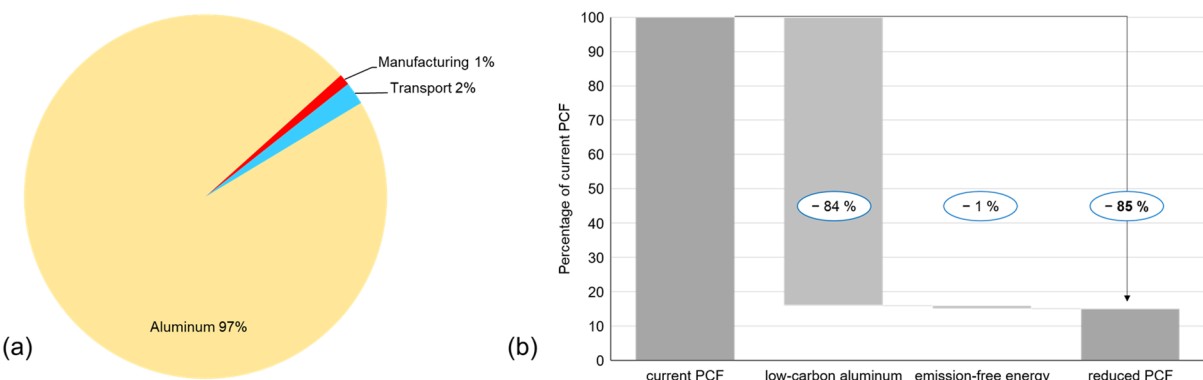

**Figure 4.** (**a**) Contribution of processes and stages to the PCF of the crossmember; (**b**) predicted GHG emission reduction potential for crossmember production.

The current PCF could be reduced by over 80% (Figure 4b). Aluminum is not only the largest contributor to PCF but also offers the greatest reduction potential. Low-carbon aluminum is made of recycled aluminum and produced with renewable energy sources, so it carries a footprint of less than 4 kg $CO_2$-eq or less per kilo of aluminum [146]. The changeover to emission-free energy has a minimal influence on the overall PCF but would eliminate almost all GHG emissions from the supplier's production. The major contribution would come from switching from gas to district heating, since the electricity purchased is already comparatively low on emissions due to the Austrian energy mix (which has a high share of hydropower) [147]. In the present case, electricity with an even higher proportion of renewable energy was intentionally sourced and was further enhanced by the installation of photovoltaic systems. The company's electricity consumption for the production of one crossmember amounted to about 90 g $CO_2$-eq in the examined time span. If the EU's energy mix were applied, the figure would be almost 169 g $CO_2$-eq (German energy mix: 256 g $CO_2$-eq; Polish energy mix: 591 g $CO_2$-eq [148]).

### 4.3. Case Study 2: Subframe

The subframe (rear-axle beam) is similar to the components analyzed by Cecchel et al. [100] and Ghosh et al. [101]. The component is composed of several parts (die-cast side parts, extruded aluminum profiles, and hydro bushings), which were manufactured by three European sub-suppliers. The final subframe consists of over 90% aluminum (alloys), with the remainder comprising steel, thermoplastics, natural rubber, and damper fluid.

In the finishing operation under consideration (Figure 5), the side parts were first welded to the profiles. This was followed by a machining process, in which approximately ten percent of the material was removed. In the final step, the bearings were pressed in. It was notable (although outside of the defined scope) that, by using pendular transportation racks (provided by OEM), the supplier did not need any packaging material for the finished subframes.

The resource consumption of the focal first-tier supplier was determined through explicit measurements and use of the manufacturing execution system. Welding consumables are especially worthy of mentioning. Intrafirm allocations followed a similar approach to that in Case Study 1. It was advantageous that the involved production line only manufactured the considered subframe. Two of the three sub-suppliers provided detailed data in the form of a PCF study or filled data collection format spreadsheets (including process flow charts). The third sub-supplier's components were evaluated on the basis of documents such as technical drawings and secondary data.

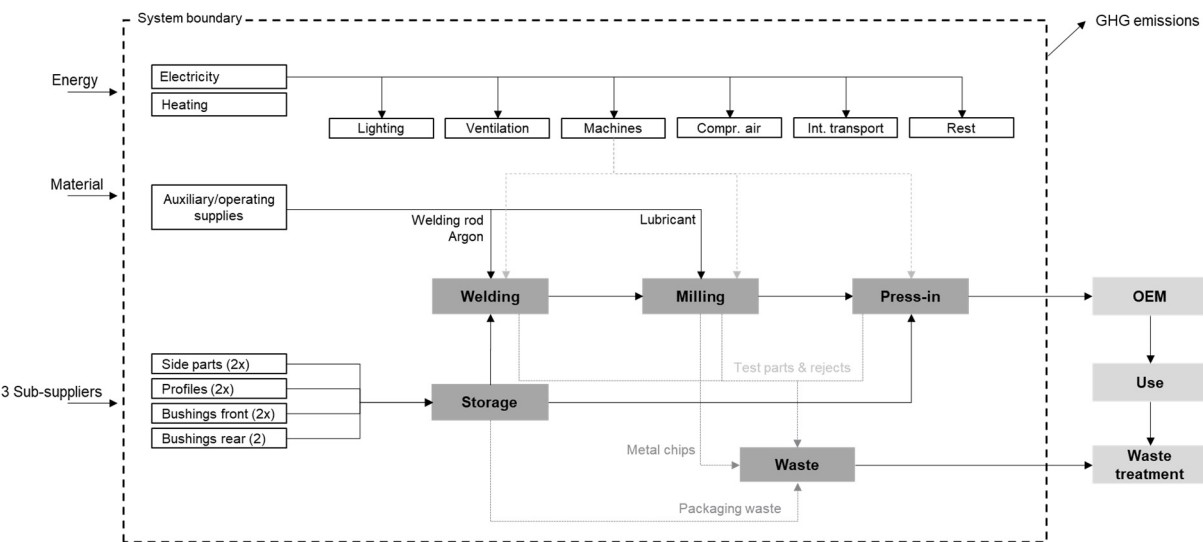

**Figure 5.** Flow chart for the subframe production.

Figure 6a shows a breakdown of the subframe's PCF. Since the side panels are the heaviest part of the subframe and were manufactured in a resource-intensive process, they accounted for 81% of total GHG emissions. The bushings accounted for ten percent of the PCF, while the profiles had a share of seven percent. The GHG emissions of the first-tier supplier were broken down as follows: milling 71%, welding and press-in operations individually 6%, and infrastructure 17%. For the sake of completeness, it is worth mentioning that a second study was carried out according to the OEM's interpretation guidelines. In this study, the contribution for raw material turned out to be even higher.

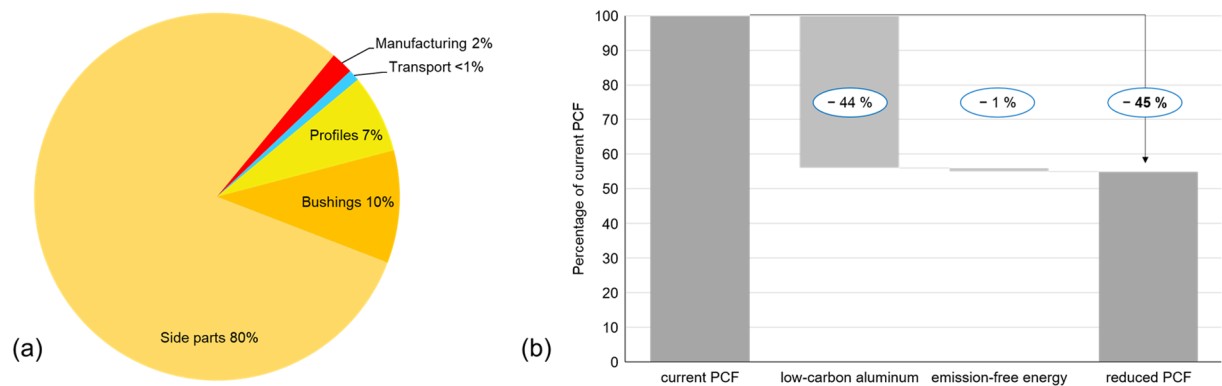

**Figure 6.** (**a**) Contribution of processes and stages to the PCF of the subframe; (**b**) predicted GHG emission reduction potential for subframe production.

Just as in Case Study 1, aluminum was the biggest driver of emissions. By switching to $CO_2$-reduced aluminum, the PCF could be reduced by 40% or more. At the first-tier supplier, purchased electricity was the biggest lever. District heating and zero-emission electricity could reduce emissions by 86% (approximately 1% of PCF). Interestingly, if the EU's energy mix was used instead of the actual electricity mix, then 7.2 kg $CO_2$-eq would be generated instead of 4.1 kg $CO_2$-eq (German energy mix: 11.7 kg $CO_2$-eq; Polish energy mix: 527.1 kg $CO_2$-eq [148]).

The scrap metal ratio of machining was higher than that reported in Cecchel et al. [100]. However, energy consumption was lower, even if the machining cycle time was longer. Therefore, the machining time (and thus power consumption) and the amount of scrap metal certainly offered further saving potentials (a fact that was not further investigated). At least the use of minimum quantity lubrication had a positive effect on the PCF. Also left out

were the optimization potentials of the product design, which could not be implemented in an economically profitable manner afterwards.

### 4.4. Case Study 3: Actuator

The considered actuator is a small drive located at the front bumper of a vehicle which performs the opening and closing movement of flaps to control the airflow to cool the engine and/or adjust the air resistance. It was produced in large quantities and delivered directly to both OEMs and first-tier suppliers. One actuator consists of 20 individual parts, 8 of which were manufactured in-house by injection molding. The complete product was made primarily of a thermoplastic polymer that was partially glass-fiber reinforced. Due to their small size and lightweight material, most components' masses were hardly measurable.

In-house manufacturing just included injection molding (but could be extended to also produce printed circuit boards (PCBs)). Purchased parts and in-house production parts were assembled through full automation. Finished products were first checked and then packed on trays before being palletized for shipment. The processes and parts are depicted in Figure 7.

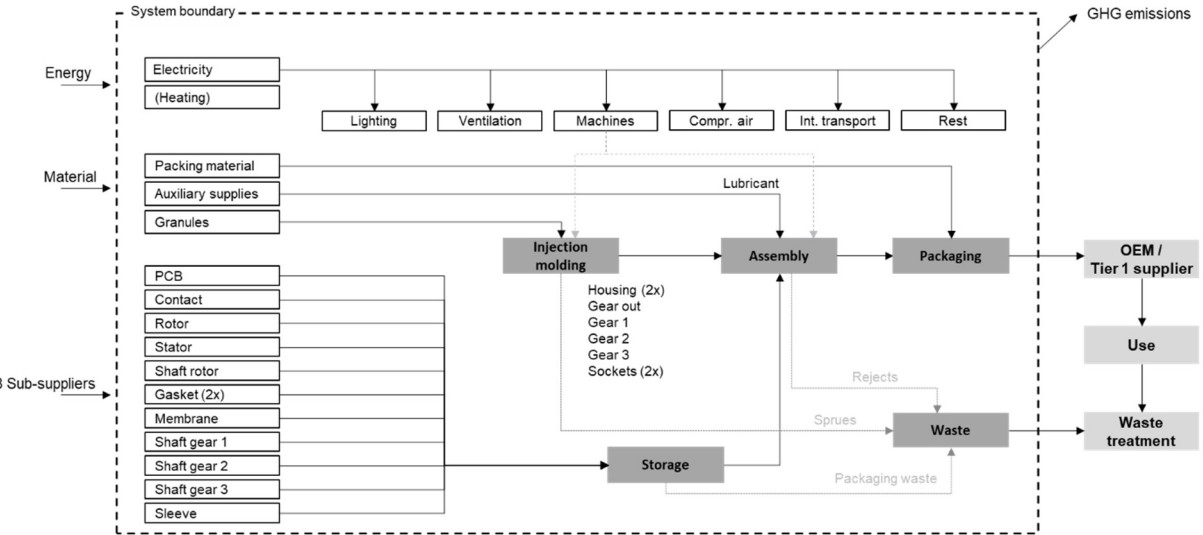

**Figure 7.** Flow chart for the actuator production.

Again, the same allocation procedures were applied as in all other cases. In this case, however, it was possible to use the number of storage spaces occupied by the product and related material instead of using a share of the storage area to allocate energy consumption for warehousing. The state-of-the-art heating system was climate neutral (i.e., no GHG emissions were detectable) and was not considered in the impact assessment.

Figure 8a portrays the distribution of GHG emissions. During the period under consideration, 55% of the total PCF was embodied in granulate material and 31% was traceable to purchased parts (for which the material input was also a major influencing factor). Because of the larger number of sub-suppliers and the longer distances (some parts were manufactured in Asia), the share of transport was found to be high in comparison to Cases 1 and 2. In-house production accounted for nine percent of the total PCF. More than half of the internal GHG emissions were caused by waste (mainly sprues and rejects), that is to say, by material once again. Approximately 20% of each was attributable to the energy consumption of actual production (manufacturing and assembly), as well as infrastructure. Once again, with regard to 17.5 g $CO_2$-eq, a comparatively low amount of GHG was emitted (EU27: 33.3 g $CO_2$-eq; Germany: 48.7 g $CO_2$-eq; Poland: 100.9 g $CO_2$-eq [148]).

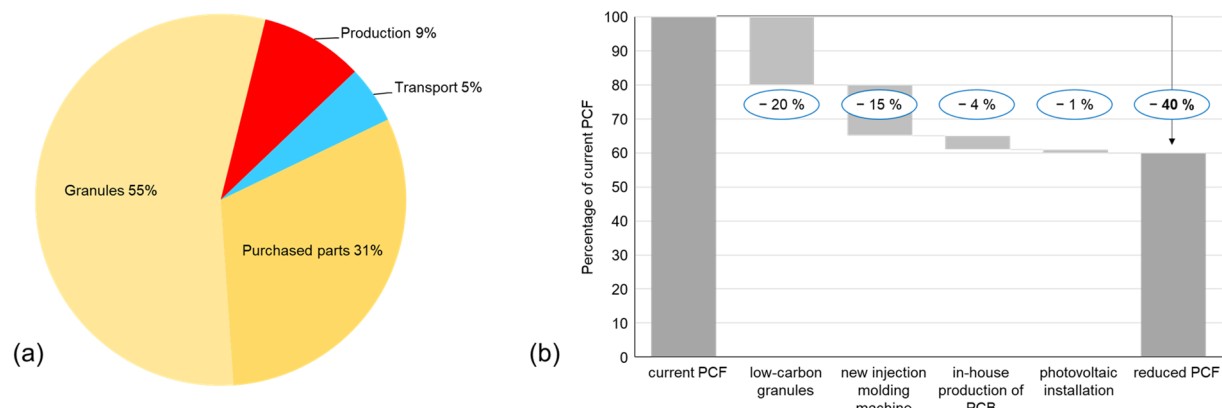

**Figure 8.** (**a**) Contribution of processes and stages to the PCF of the actuator; (**b**) predicted GHG emission reduction potential for actuator production.

At least for some parts, the glass fiber-reinforced polyamides can be replaced by other plastics. Suitable materials include carbon fiber-reinforced polypropylene and bio-based materials (as demonstrated, for example, by [134]). Rough estimates indicate a savings potential of 20%. The planned replacement of the injection molding machines to ones that feature direct injection and vacuum pumps will significantly increase efficiency. Considerably less energy (both electricity and compressed air) is consumed while, simultaneously, material is saved by avoiding sprues. Shifting PCBs from external to internal production will save energy and reduce the amount of packaging waste. Finally, an intended photovoltaic system can cover one third of the site's electricity demands. While this only slightly reduces the total PCF, it nonetheless results in a significant reduction to the corporate carbon footprint.

*4.5. Comparison of Cases*

All three of the previously described case studies have the following in common: they examine direct and large-scale production (the annual output volume of several tens of thousands of units) in Austria and verify that the upstream activities (i.e., raw and precursor materials) are the largest contributor of an Austrian automotive supplier's PCF. There are two reasons why the considered in-house production contributed only a single-digit percentage to the PCF. First, the analyzed factories are up to modern (energy) standards. Second, Austria has an energy mix with a very low GHG emission intensity [148]. Therefore, in two of the three case studies, building infrastructure (HVAC and lighting) accounts for only a small share of GHG emissions. Dimensions, as reported by Gebler et al. [97] and Sullivan et al. [52], could not be observed (at least not when broken down to the product level).

Table 3 compares the three components based on product complexity, proportion of in-house production, the key drivers of the PCF, and the internal PCF reduction potential, as well as the PCF reduction potential of sourcing at the overall product level. Except for the key drivers, an ordinal scale was used for categorization.

**Table 3.** Cross-case analysis.

| Component | Complexity | Level of In-House Production | Key Drivers | Internal Potential | Potential via Sourcing |
|---|---|---|---|---|---|
| crossmember | very low | very low | material | low | very high |
| subframe | low | low | material | low | high |
| actuator | medium | medium | material, production technology | medium | medium |

The crossmember is a very simple product that requires only one production step in-house. However, the press involved is also used for the production of other components, which complicates the measurement, calculation, and allocation of input and output. Although the subframe consists of several parts and requires more processing steps, which increases the product complexity, the data collection and allocation is simple, since the production line only manufactures this component. Due to the high number of parts and sub-suppliers involved, the actuator has a higher complexity. In addition, there is a considerable proportion of in-house production, which makes the machinery used to another important driver and lever of the PCF.

With regard to optimization potential, two factors are always considered: an efficient use of sustainably produced materials and the switch to emission-free energy. Both aspects have already been covered in the literature (e.g., [104]). From a production perspective, changing to a secondary or even bio-based material is easy and fast to implement since (usually) no adjustments to the machines are necessary. However, the experience from these three cases showed that the high requirements placed on the materials (e.g., weldability, machinability, and durability, as well as strength and elasticity) and the concerns and formalities of the OEMs complicated the change of materials. No less crucial is the OEM's willingness to pay for low-carbon components. Although economic aspects such as surcharges were intentionally ignored in the case studies, interviewed stakeholders mentioned that the vehicle sector is so competitive and its margins so low [149] that OEMs would generally not accept extra costs. Moreover, sustainable manufacturing as a unique selling proposition may also not allow for the passing on of additional costs to the customer [150]. On the other hand, some research [151–153] predict that there will be little additional cost if a vehicle is made from low-carbon materials, which makes our recommendations for a transition in sourcing more feasible. However, a basic prerequisite is that the demand for this material can be met (e.g., through sufficient recycling rates (see the prognosis of the International Aluminium Institute) [154]).

### 4.6. Discussion and Implications of Findings

All case studies show that industrial manufacturing in Austria emits comparatively little GHG and contributes only a small part to the PCF. The results suggest that, for simple components with little in-house production, sourcing has particularly large potential. As the proportion of in-house production increases, internal PCF reduction potentials become more important. Our findings resonate with the literature regarding environmentally responsible procurement [59,60]. They underline the importance of sourcing low-emission materials and its GHG-saving potential at the product level. However, in contrast to the majority of the publications mentioned in Section 3, the PCF reduction potential identified in the cases did not result from fundamental different materials or manufacturing processes but from quickly implementable adjustments.

The presented approach offers basic guidance when manufacturers want to identify and evaluate their product's impact and associated reduction potentials. Management can not only learn from this for upcoming requirements by legislation and OEMs but can also become aware of drivers and (further) reduction potentials of PCFs. Our study also provides implications for academia, as it addresses the research gap regarding PCF related to Austrian industry.

The predicted GHG emission reduction potentials have at least two limitations. Firstly, in only one of the three studies we considered the best available technology, and, even then, no exhaustive technology screening was conducted. Associated probability analyses (especially calculations on payback periods) were not made. Furthermore, measures to increase efficiency, e.g., through optimized machine settings, were not assessed. Secondly, our studies were by nature limited to a specific time period. A sensitivity analysis that shows how PCFs and potentials depend on production volume and capacity utilization is still to be conducted.

A further, yet difficult to quantify, finding was the observed 'polluter pays principle'. It could discourage suppliers from investing in low-carbon designs, processes, and materials since they will not receive any benefits for being proactive.

## 5. Conclusions and Outlook

One third of GHG emissions are attributable to industry and, contrary to other sectors, have hardly improved in recent years. The climate targets of developed countries such as Austria are important. However, it must not be forgotten that production (especially raw material extraction and the manufacturing of parts) has been outsourced to countries that produce more at lower costs but not necessarily at lower GHG emissions. Besides upcoming EU regulations, we brought up the net-zero pledges of automobile manufacturers. For a long time, the automotive industry has largely focused on reducing tailpipe emissions, but with breakthroughs in electric vehicles, the balance of the sector's carbon footprint is shifting to ecological vehicle production.

Since a vehicle uses resources and generates emissions before ever having driven its first kilometer, LCAs are used to determine the total harmful impact on the environment. By doing so, the true extent of the environmental load caused by the automotive industry becomes clear: it consumes one fifth of the world's aluminum and steel production and large quantities of other materials. This paper demonstrates that the majority of a vehicle production's 'carbon rucksack' can be traced back to suppliers and even further back to activities close to the 'cradle'. The success of OEMs' responses to climate change is therefore heavily dependent on the performance of their supply chain and the sourcing within.

Although numerous LCAs and PCF studies of vehicle components have been published in recent years, there is a lack of comparable studies with production sites in Austria. This paper is the first step in expanding the existing literature and addressing the research gap. Our three cases could demonstrate that the embodied emissions of (imported) materials affect the PCF much more than the finishing carried out by the supplier companies located in Austria. In all case studies, the final production stage(s) accounted for less than ten percent of the total PCF. In brief, we conclude the following: first, the substitution of current materials with low-carbon materials (i.e., recycled, bio-based, or renewable energy produced materials) is the most effective lever for reducing PCFs and assigns great responsibility to procurement. From the present project experience, we also conclude that the responsibility for ecologically sustainable production has been passed from OEMs to the suppliers. Second, within a factory, electricity is of particular importance. Even in a country such as Austria, a switch to emission-free energy sources will accomplish a significant effect. Thirdly, to be able to conduct PCF studies, there is still a need for qualification.

As a practical contribution, our findings can be used by companies to prepare for future PCF reporting. The illustrated reduction potentials have been perceived as useful (in achieving potential market advantages) by representatives of the involved companies, especially if PCFs are to become a sourcing decision criterion. In analogy to Lee (2011) and to ensure (future) applicability of our approach, step-by step guidelines and individual templates for recording resource consumption have been developed for all cases, with a particular focus on assisting allocation procedures. In addition, the experience gained will not only support the companies involved but will also be used in the engineering education at Graz University of Technology. We believe it is of particular importance to raise awareness of the relevance of upstream activities as well as the challenges in defining system boundaries and allocating resource consumptions.

Based on our findings, future research should consider additional cases and focus on aspects outside the analyzed scope, such as the excluded engineering activities, best available technologies, or economic viability of decarbonization measures. Additionally, further research is required to address technical limitations such as the limited availability of sustainably produced materials.

**Author Contributions:** Conceptualization, M.W.; Investigation, K.R. and M.W.; Validation, K.R.; Writing—original draft, K.R.; Writing—review & editing, M.W.; Visualization, K.R. and M.W.; Supervision, M.W. All authors have read and agreed to the published version of the manuscript.

**Funding:** This publication was supported by the Open Access Funding by the Graz University of Technology.

**Institutional Review Board Statement:** Not applicable.

**Informed Consent Statement:** Not applicable.

**Data Availability Statement:** The data used to support the findings of this study are available from the corresponding author upon request.

**Acknowledgments:** The authors would like to thank the representatives of both companies for authorizing the use of data for scientific purposes. We would also like to thank Sila Temizel-Sekeryan and Manuel Ulrich for their invaluable support.

**Conflicts of Interest:** The authors declare that they have no known competing financial interest or personal relationship that could have appeared to influence this article. We also did not receive any specific grant from funding agencies in the public, commercial, or not-for-profit sectors.

## Abbreviations

| | |
|---|---|
| ETS | Emissions Trading System |
| EU | European Union |
| GHG | greenhouse gas |
| GRI | Global Reporting Initiative |
| GVA | gross value added |
| HVAC | heating, ventilation, air conditioning |
| IPCC | Intergovernmental Panel on Climate Change |
| ISIC | International Standard Industrial Classification (of all economic activities) |
| LCA | life cycle assessment |
| LCI | life cycle inventory (analysis) |
| OECD | Organization for Economic Co-operation and Development |
| OEM | original equipment manufacturer |
| PCF | product carbon footprint |
| VDA | German Association of the Automotive Industry (Verband der Automobilindustrie) |

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
