# Peer review of "Identification and Reduction of Product Carbon Footprints: Case Studies from the Austrian Automotive Supplier Industry"

_sustainability, doi:10.3390/su152014911_

Round 1

Reviewer 1 Report

The case study in this research attempts to identify involved factors/entities contobuting the emission/GHS. Some fedbacks to upgrade the quality of the paper before publications:

1. Abstract should be revised significantly in term of the methods used in this study.

2. Introduction should be focused on the prior related works to lead the novelty or gaps of this study compared to them.

4. please consider to add one or two grounded theory in the section of literature review.

5. There is no research methodology in the paper, so author should add the section.

6. Please make separation between results and discusssion in the paper. Discussion would be the most important part of the sections in the paper.

7. Together with discussion section, auhtors should also consider add the additional section, such as managerial implications and theoritical contribution for scholars/academics. Managerial implications are for the stakeholders involved in the GHS reduction such as the government, company, community or unions. Theoritical contributions would be relevent with the novelty/gaps mentionded in the introduction.

Reviewer 2 Report

The manuscript seems like a scientific report, not a research article. There are no formulas or mathematical models to support the results. The innovation of this paper is insufficient.

Reviewer 3 Report

Abstract:

Need to highlight the method clearly and also contribution to knowledge as well. 

Introduction

You need to highlight in the introduction why the three case studies are important, what is the other study's limitation regarding to the topic.

Clarify the research gap in the introduction and please describe how your achievement to fill the gap in the discussion section

Case study 1 2 3

Please make a table comparison to serve easier understanding for the reader. what are pros and cons of each case study.

Where is your methodology, need to add the section of methodology?

Need to highlight the method clearly and also contribution to knowledge as well. 

Discussion

Need deeper discussion, especially the comparison of the existing study with your case study result.

Conclusion

Sufficient

Round 2

Reviewer 2 Report

The manuscript has been revisied well.